# A High-Temperature Multipoint Hydrogen Sensor Using an Intrinsic Fabry–Perot Interferometer in Optical Fiber

Rongtao Cao [1], Jingyu Wu [1], Yang Yang [2], Mohan Wang [1], Yuqi Li [1] and Kevin P. Chen [1,*]

1   Electrical and Computer Engineering Department, University of Pittsburgh, Pittsburgh, PA 15261, USA
2   School of Optoelectronic Engineering and Instrument Science, Dalian University of Technology, Dalian 116081, China
*   Correspondence: pec9@pitt.edu

**Abstract:** This paper presents a multiplexable fiber optic chemical sensor with the capability of monitoring hydrogen gas concentration at high temperatures up to 750 °C. The Pd-nanoparticle infused $TiO_2$ films coated on intrinsic Fabry–Perot interferometer (IFPI) array were used as sensory films. Strains induced upon exposure to hydrogen with varied concentrations can be monitored by IFPI sensors. The fiber sensor shows a repetitive and reversible response when exposed to a low level (1–6%) of hydrogen gas. Uniform sensory behavior across all the sensing cavities is demonstrated and reported in this paper.

**Keywords:** optical fibers; hydrogen sensor; Fabry–Perot interferometer; high temperature

## 1. Introduction

Hydrogen fuels have been considered as a possible clean energy source to drive a carbon-free global economy. The consumption of hydrogen for electricity production or transportation does not generate carbon dioxide. Unlike fossil energy feedstock, which is only available in limited locations on Earth, the distribution and local production of hydrogen using renewable energy is possible to improve sustainable economic developments and energy securities. However, large-scale utilizations of hydrogen as fuels incur significant safety concerns. As a highly reactive molecule, hydrogen gas has a low explosion limit (4 vol%) and low ignition energy (0.02 mJ) in air. The small molecule size of hydrogen is also prone to leak from the transportation networks and storage facilities. This produces significant safety concerns to facilities and personnel. To better address this challenge, both industry and academic researchers have been developing low-cost hydrogen sensors that can perform rapid and accurate measurements of hydrogen with reasonable sensitivities. The large-scale deployment of hydrogen sensors can effectively mitigate safety concern and improve the prospect of the hydrogen economy if they can be deployed economically at low costs. Hydrogen gas sensors based on optical fiber platforms have been considered as a very promising solution for the hydrogen economy due to the small form-factor, high sensitivity, and distributed sensing capability. Multiple optical hydrogen sensor prototypes have been reported based on various fiber optic sensing scheme such as evanescent-wave absorption [1,2] and fiber Bragg-Grating [3,4]. Especially made fibers such as tapered fiber [5] and microfiber [6] could also improve the sensing sensitivity, and have been utilized for hydrogen sensing [7]. However, these existing sensors are designed to operate at room temperatures or low temperatures. As the majority of hydrogen gas is consumed through various high-temperature processes such as combustion or through solid oxide fuel cells, there is no reported multiplexable hydrogen sensor that can perform high spatial resolution measurement to directly monitor the energy production process at high temperatures. Sapphire fiber has been used to perform single-point hydrogen sensing at high temperatures over 800 °C with a metal oxide functional coating [8]. However, sapphire fibers are intrinsically multimode fibers, and this, together with their high-cost, limits the application

of sapphire fibers as effective sensor platforms to perform multiple-point hydrogen sensing. Another issue related to passive sensing devices such as the fiber Bragg Grating sensor is that it requires high-cost wide band optical spectrum analyzers with high-resolution. It also requires a complicated optical setup for grating inscriptions such as the phase mask and ultraviolet laser [3,4,9]. Hydrogen sensors based on extrinsic Fabry–Perot interferometers have also been demonstrated by many previous efforts with high resolution and low cost [10–12]. However, most of those works are utilized for monitoring the hydrogen gas at low temperature. The underlying reason is that the sandwiched structures of extrinsic Fabry–Perot make them impossible to be functional at high temperature applications due to the stability issues of the cavity.

In this paper, we demonstrate multiplexable fiber optical hydrogen sensors based on low-cost silica fibers to achieve operational temperatures as high as 750 °C. Multiple high-temperature stable intrinsic Fabry–Perot interferometers (IFPI) were inscribed in the optical fiber core as strain sensors through a femtosecond laser writing process. Palladium (Pd) infused metal oxide (Titanium dioxide ($TiO_2$)) was prepared through a sol-gel synthesis scheme and utilized as the functional coating material. The expansion of the metal oxide thin film during the formation of palladium hydride upon hydrogen exposure generated a tensile strain being transferred and monitored by the IFPI cavities. A hydrogen sensor with good repeatability and reversibility when exposed to gas levels above 1% is reported in this paper. The effects of film thickness and the long-term stability of the sensory films at high temperatures are studied in this paper. The multiplexability of these fiber sensors are demonstrated by a string of three IFPI hydrogen sensors inscribed on the same fiber, which exhibit a consistent and repeatable hydrogen response from 1% to 6% hydrogen concentration. The result reported in this paper suggests that the integration of functional sensory film on multiplexable and highly sensitive fiber optical sensor devices can be a viable approach to perform high spatial resolution hydrogen measurements for high temperature environments.

## 2. Materials and Methods

The sensing cavities were fabricated on a standard single mode fiber (Corning SMF-28e+, Glendale, AZ, USA). As shown in Figure 1a, a linearly polarized laser beam from a coherent RegA 9000 Ti:sapphire laser system (Coherent, Inc., Santa Clara, CA, USA) was focused inside the center of the SMF core to generate the Rayleigh scattering defects [13–15]. An oil-immersion lens with a numerical aperture of 1.25 and magnification of 100× was introduced to produce the tight focus. The laser beam with the wavelength of 800 nm was optimized to be 270-fs at a 250-kHz repetition rate. Using this fabrication approach, up to 20 high-temperature stable IFPI sensors were successfully fabricated [13]. The Pd-infused metal oxide functional coating was prepared through a sol-gel synthesis approach [8]. The metal oxide precursor was formed by mixing 1.5 g $Ti(OCH(CH_3)_2)_4$, 0.45 g HCl and 6 g ethanol (ACS Reagent Grade, Sigma Aldrich, Burlington, VT, USA). Then, 0.06 g $PdCl_2$ (Sigma Aldrich) was added to 5.6 g ethanol, followed by 0.45 g $HCl_2$, to form the Pd mixture. One gram of the Pd mixture was added to the prepared metal oxide precursor and stirred for 10 min to form a uniform solution. Then, 0.8g Pluronic F-127 (Sigma Aldrich) block copolymer template was added to the uniform mixture solution for the formation of a 3D nanostructure. The solution was ready to be used after 12 h of stirring at 60 °C. The dip coating method was utilized in the experiment to form a thin layer of Pd functionalized $TiO_2$ coating on top of the laser processed SMF. The coated fiber was then transferred into a tube furnace and underwent a calcination process at 600 °C for 1 h. A sensor with a single cavity of 458.32 µm was first fabricated in the experiment. Figure 1b shows the cross-sectional view of the prepared sensor by Scanning Electron Microscope (SEM). There were small variations in coating thickness, and the average coating thickness of the sensor was around 8 µm. The thickness of the coating was controlled by the coating time. The sol-gel film with the addition of Pluronic F-127 produced nanoporous structures with the grain size ranging from 50 nm to 100 nm, as shown in the inset of Figure 1b.

Pd nanoparticles were formed inside nanoporous $TiO_2$ with sizes of ~20 nm, which was confirmed by previous Transmission Electron Microscopy studies [8]. The porous structures in the sensory film drastically enhanced the hydrogen gas interaction by increasing the surface-to-volume ratio.

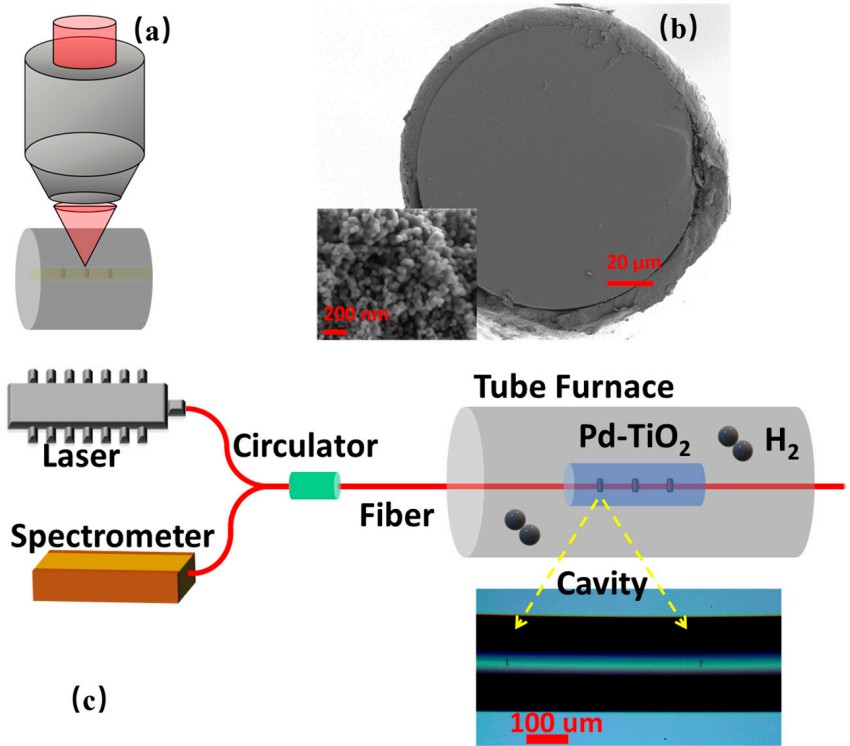

**Figure 1.** (**a**) Schematic of sensing cavity fabrication through laser processing. (**b**) Cross-sectional view of the fabricated sensor; inset shows the enlarged view of the sensory film. (**c**) Schematic of the experimental setup; inset shows the optical scope image of the sensing cavity after laser-processing.

Figure 1c shows the schematic diagram of the experimental setup. A broadband source (Exalos) with a tunable range of 60 nm at the center wavelength of 1550 nm was introduced as the light source during the experiments. Hydrogen gas with varied concentrations was delivered into the tube furnace where the fiber optics sensor under test was located. The gas flow of hydrogen and nitrogen was fixed to be 20 sccm. The operating temperature of the tube furnace was settled to be 750 °C. The backscatter light was collected by the spectrometer (BaySpec) through a circulator. The inset shows the fabricated Fabry–Perot cavity with two reflective surfaces through laser processing.

## 3. Results

The interference spectrum with fringes ranging from 1505 nm to 1590 nm was recorded by the spectrometer, as shown in Figure 2a. The demodulated cavity length was 458.32 μm, as plotted in Figure 2b. Fast Fourier transform was implemented together with Buneman frequency estimation to perform the demodulation process and the experimental demodulation resolution of the cavity length was 64 nε, corresponding to 0.2 nm in cavity length [16,17]. The sensory film thickness, measured by SEM, was estimated to be around 8 μm. The response of the sensor upon gas exposure at high temperature is shown in Figure 2c. An increasing tensile strain was observed when exposed to hydrogen gas. The tensile strain ranged from 18 με to 50 με after 15 min exposure with the hydrogen concentration between 1% and 6%. The tensile strain expansion recorded by the IFPI sensor was the result of the Pd lattice constant expansion in the sensing coating upon the formation of palladium hydride during gas exposure [18,19]. It is worth noting that the tensile strain kept increasing when the hydrogen gas was being continuously delivered. No mechanical

delamination between sensory film and silica fibers was observed. This is probably due to the use of nanoporous sensory film. As we reported before [20–22], both optic (refractive indices) and chemical parameters (surface-to-volume ratio) can be tuned by the porosity of the functional film. This paper also reveals that mechanic properties (Young's modulus) can also be tuned to improve the adhesion of the sensory film. The reversibility of the sensor is also worth investigating. In this paper, nitrogen gas was introduced for 25 min after hydrogen exposure to reset the sensor. As shown in Figure 2c, the tensile strain drops and resets back to its initial level after the nitrogen exposure, indicating the dissociation of the palladium hydride and confirming the reversibility of the fabricated sensor. The temperature fluctuation of the tube furnace could also generate potential cavity length and tensile strain changes that introduce cross-sensitivity. To study the influence of the tube furnace temperature stability on the measurement accuracy, no gases were introduced to the tube furnace for 1 h at 750 °C and the time-dependent cavity length change was recorded as shown in Figure 2d. The maximum cavity length variation was less than 3 nm, corresponding to less than 6 με. It was less than one third of the tensile strain introduced by exposure to 1% of hydrogen gas, and thus has very little effect on the measurement accuracy. This result matches our previous study of an IFPI sensor at high temperature, as the measured temperature fluctuation is less than 0.5 °C by thermocouple and the corresponding temperature sensitivity was calculated to be 5.6 nm/°C [23]. In a future study, a reference sensor could be introduced into the tube furnace along with the hydrogen sensor to record the cavity length variation due to temperature fluctuation and compensate it from the hydrogen sensor to improve the measurement accuracy.

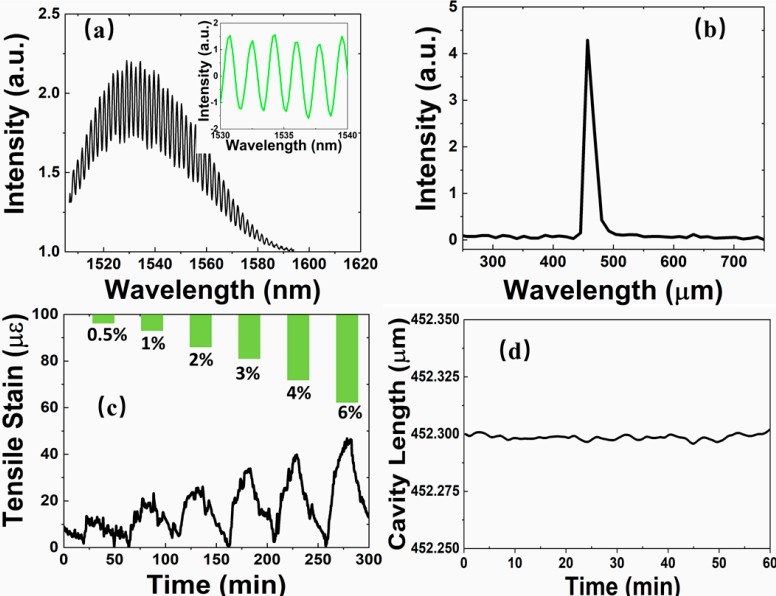

**Figure 2.** (**a**) Interference spectrum of the cavity recorded by spectrometer, and the inset shows the interference spectrum after the light source baseline profile is removed. (**b**) Demodulated cavity length. (**c**) Time-dependent response of the sensor to hydrogen exposure with various gas concentrations. (**d**) Time-dependent response of the sensor under no gas exposure for 1 h.

Since the IFPI sensor monitors tensile strain change introduced by the functional film as the indicator of hydrogen concentration, the coating thickness of the film plays a significant role in determining the performance of the sensors. To investigate the impact of sensory film, a hydrogen sensor with the functional coating thickness reduced to 3 μm on average was fabricated. The sensor with thinner coating layer was exposed to the hydrogen gas with the same delivery sequence and exposure times for testing, and its cavity length change rate is plotted in Figure 3. In the figure, compared with thick film coatings of 8 μm, the sensor with the thinner coating generates a less significant tensile strain change. For

instance, the strain change is 42 µε for sensor with the thin coating layer when exposed to 6% of hydrogen gas, 5% less than the response of the sensor with thick coatings of 8 µm. The discrepancy gradually increases with the reducing gas concentrations. For example, compared with a thick functional sensing layer, a sensor with a thinner coating exposed to 1% of hydrogen gas introduced a tensile strain change 33% less than that of the thick coating one. One possible reason for the increased difference between sensors with varied coating thicknesses at lower gas level might result from the increased surface area of the thick coating that accommodates more Pd nanoparticles for gas absorption, as well as the unsaturated palladium hydride formation process.

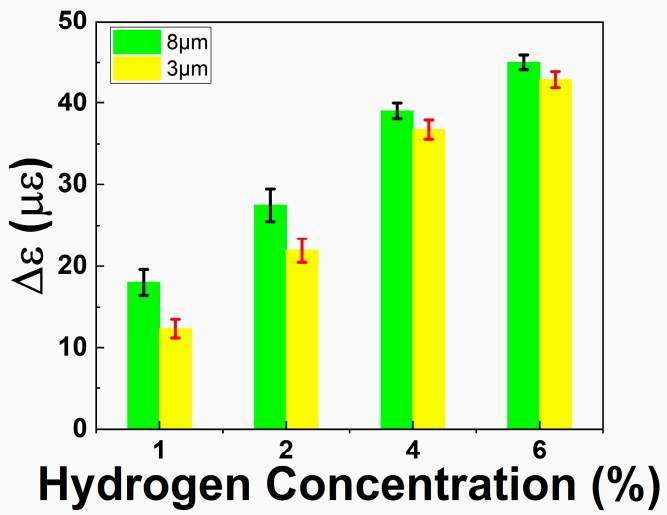

**Figure 3.** Hydrogen-induced tensile strain changes for the sensor with 3-µm thick functional coating layer vs. sensor with 8-µm thick coating. The error bars are calculated according to standard deviation formula [24].

One significant advantage of using IFPI devices as sensor platforms is their high degree of multiplexability. Following the study of the single IFPI sensor, a sensor with three IFPI sensing cavities was also fabricated and studied in the experiment. The interference pattern and demodulated cavity length of three IFPI sensors with cavity lengths of 429.14 µm, 964.65 µm and 1476.19 µm are shown in Figure 4a,b, respectively. During the sensor characterization process, multiple cycles of hydrogen gas ranging from 6% to 1% were delivered into the tube furnace, followed by nitrogen gas, resetting the process after each hydrogen gas exposure. The gas flux was also fixed at 20 sccm, and the furnace was heated to 750 °C. This was identical to the process used to study the single IFPI sensor device.

The time dependent tensile strain response of the three sensing cavities to the gas concentration variations is shown in Figure 4c, which shows repeatable responses when exposed to multiple cycles of the hydrogen gas. All three sensors show consistent hydrogen responses, which exhibit similar hydrogen responses, as shown in Figure 4d. For individual sensors, repeated cycles of hydrogen exposure with the same concentration resulted in almost identical strain responses. For example, the tensile strain variation for the IFPI sensor with the cavity length of 429.14 µm reduced from 4% to less than 2.5% when the hydrogen gas concentration increased from 1% to 6%. The hydrogen-induced tensile strain variations among three IFPI devices were also highly consistent. The variation of averaged strain change was 1.5 µε for 1% hydrogen exposure, and this variation decreased to 0.86 µε for 6% hydrogen among three sensing devices.

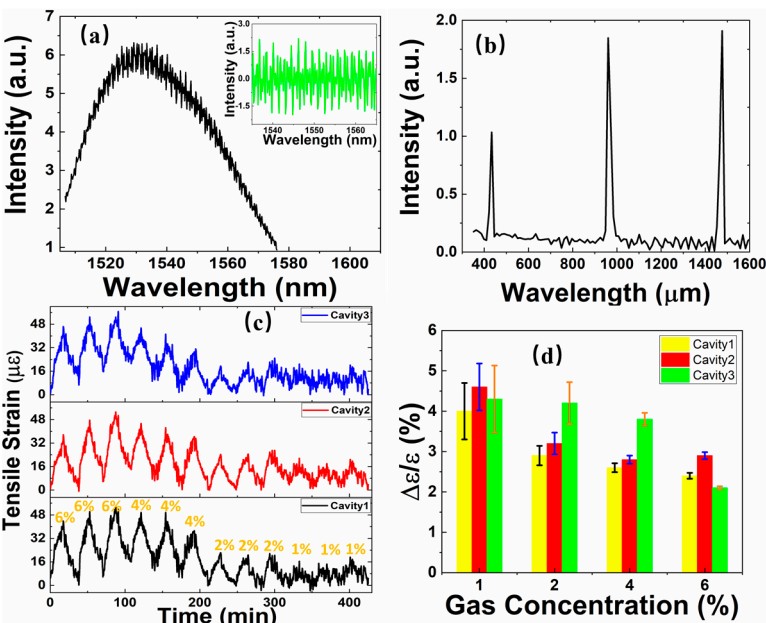

**Figure 4.** (**a**) Interference spectrum of the IFPI sensor with three sensing cavities. The inset shows an interference spectrum after the light source baseline profile was removed. (**b**) Demodulated cavity length distribution. (**c**) Time-dependent response of IFPI sensors to hydrogen exposure with concentration ranging from 6% to 1%. (**d**) Tensile strain change variations across the multiple gas exposure cycles and sensing cavities. The error bars are calculated according to standard deviation formula [24].

## 4. Discussion and Conclusions

This paper demonstrates a multipoint hydrogen sensor based on a highly multiplexable intrinsic Fabry–Perot interferometer for high temperature chemical sensing applications. Pd-doped metal oxide was utilized as the functional coating material. Multiple IFPI cavities inscribed to the optical fiber were utilized to monitor the tensile strain variation of the coating film upon hydrogen exposure, with different concentrations at high temperatures. A repeatable and reversible response was also observed across the multiple cycles of gas exposure, as well as the different sensing cavities at 750 °C. In this paper, the hydrogen sensing experiments were performed for hydrogen concentrations between 1% and 6%, which was limited by our gas supply setup. However, it was noted that 1% hydrogen induced significant axial strain of 18 µε on the fiber. Using the Buneman frequency estimation, it was possible for IFPI to detect much smaller static strains of 7 nε [25]. This suggests that the multiplexed IFPI hydrogen sensor demonstrated in this paper can detect hydrogen at much lower concentrations.

Many technical challenges still remain to bring fiber optical hydrogen sensors from the laboratory to industry applications. However, this paper shows that the use of highly multiplexable fiber optical hydrogen sensors provides a very appealing and unique sensing solution for large-scale hydrogen infrastructures monitoring at every stage of the hydrogen value chain. The ability to deploy multiple hydrogen sensors interrogated by a single sensor reading system enables the utility operators to harness high spatial resolution information to cover entire hydrogen facilities, which will significantly improve their operational safety. The integration of IFPI devices and a wide array of other sensory films could also yield a new class of multiplexable fiber optical chemical sensors.

**Author Contributions:** Conceptualization, K.P.C. and R.C.; methodology, R.C.; software, Y.Y. and Y.L.; validation, R.C., J.W., M.W. and K.P.C.; formal analysis, R.C.; writing—review and editing, R.C. and K.P.C. All authors have read and agreed to the published version of the manuscript.

**Funding:** U.S. Department of Energy (DE-FE0032210).

**Institutional Review Board Statement:** Not applicable.

**Informed Consent Statement:** Not applicable.

**Data Availability Statement:** The data presented in this study are available upon reasonable request from the corresponding author.

**Conflicts of Interest:** The authors declare no conflict of interest.

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
