# Peer review of "A High-Temperature Multipoint Hydrogen Sensor Using an Intrinsic Fabry–Perot Interferometer in Optical Fiber"

_photonics, doi:10.3390/photonics10030284_

Round 1
Reviewer 1 Report
This paper proposed a Hydrogen sensor based on the combination of Pd doped metal oxide film and femotosecond laser written IFPI structure. The sensor can work in a high temperature condition up to 750℃, and sensor multiplexing can be easily realized. Overall, the paper is written with a high quality. The performances of the proposed sensor have been fully studied under a high temperature. Thus, I am pleasure to recommond its publication on Photonics. However, a few concerns need to be fully addressed before its final acceptance. My detailed comments are as follows:
1) A microscopic hotograph of the FP structure after fs written needed to be given in Figure 1 or other places.
2) The cavity length demodulation accuracy based on the FFT anddbuneman frequency estimation should be given for the estimation of its hydrogen sensing resolution.
3) The sensor was experimentally characterized under a high temperature of 750℃, however, in real cases, the sensor will not work in a stable temperature. The temperature is generally fluctuated with time. As temperature will also affect FP’s cavity length, under different temperature, the cavity length will be very different, obviously, there exist a temperature cross-sensitivity of temperature on hygrogen concentration. So the temperature response of the sensor need to be given. And the authors needed to tell the readers how to compensate or solve the temperature cross-sensitivity.
Reviewer 2 Report
The authors demonstrate multipoint intrinsic Fabry-Perot interferometer for hydrogen sensor. The paper has some interest for the field of high-sensitivity optical fiber sensing. The manuscript can be accepted after responding the following questions.
1) The cavity length of IFPI has a few hundred micrometers, while the film thickness has only several micrometers. Therefore, the variation of film thickness would be neglect relative to that of IFPI. The sensitivity would be relatively low.
2) Why are the lateral surface and end surface of optical fiber simultaneously coated by the film, considering optical light just propagation in the fiber core? Is there some strain of silica fiber because of the expansion of lateral film? However, silica fiber has large Young’s modulus, and is difficult to be stretched. Please give some more explanation and description.
3) Why is the film thickness chosen 3 micrometer in the case of multiple sensors?
4) In Fig. 4(c), there are not distinct change regulation in the case of three sensor as increasing gas concentration. Is it this reason to choose another variable of tensile strain?
5) Three sensors has the gradually increasing cavity lengths, while there is few variation regulation in the variable of tensile strain as shown in Fig. 4(d), as increasing gas concentration. Please give some more detailed explanation.
6) I suggest adding at the Authors’ reference of the recent reference about the high-sensitivity fiber sensing field of “Optics Letters, 45(14), 3889-3892, 2020”. In addition, coating film on the surface of tapered fiber for gas sensing would be a good potential.
Round 2
Reviewer 1 Report
All my concerns have been responsed properly. I think it can be accepted in tis current form.
Author Response
Thank you for your comments. We do appreciate your time and the useful suggestions.
Reviewer 2 Report
Thank the authors to give the detailed response. I have no comment, and the manuscript can be accepted and published in the journal.
Author Response

(The authors gave the same response as above.)

Reviewer 3 Report
The manuscript still needs some improvement, but should be publishable after the flowing minor issues are addressed:
Line 15: “when expose to low level (0.5%) of hydrogen gas”. This should be revised. According to the results, the sensor also shows reversible response when exposed to higher levels of hydrogen.
Line 68: “detection limit of 0.5%”. 0.5% was the lowest hydrogen concentration tested. However, the detection limit was not calculated. According to the results shown in Figure 4(c), the signal is very noisy even at 1% hydrogen. So, the detection limit is estimated to be higher.
Line 141: “significantly larger than the system’s detection limit”. Because the detection limit of the system was not calculated and provided, this sentence should be removed or the detection limit can be calculated and provided here.
Figure 3 and Figure 4(d): How the error bars were calculated should be explained in the figure legends.
